# Induction of Sustained Immunity Following Vaccination with Live Attenuated *Trypanosoma cruzi* Parasites Combined with Saponin-Based Adjuvants

**DOI:** 10.3390/biology14091298

**Published:** 2025-09-20

**Authors:** Brenda A. Zabala, María Elisa Vázquez, Daniela E. Barraza, Andrea C. Mesías, Federico Ramos, Alejandro Uncos, Iván S. Marcipar, Leonardo Acuña, Cecilia Pérez Brandán

**Affiliations:** 1Unidad de Biotecnología y Protozoarios (UBIPRO), Instituto de Patología Experimental “Dr. Miguel Ángel Basombrío”, Consejo Nacional de Investigaciones Científicas y Técnicas (CONICET), Universidad Nacional de Salta, Salta A4400, Argentina; b.zabala@conicet.gov.ar (B.A.Z.); elisa.vazquez@conicet.gov.ar (M.E.V.); barrazadaniela@conicet.gov.ar (D.E.B.); 2Instituto de Patología Experimental “Dr. Miguel Ángel Basombrío”, Consejo Nacional de Investigaciones Científicas y Técnicas (CONICET), Universidad Nacional de Salta, Salta A4400, Argentina; amesias@conicet.gov.ar (A.C.M.); federamosqui@gmail.com (F.R.); auncos@gmail.com (A.U.); 3Facultad de Bioquímica y Ciencias Biológicas, Universidad Nacional del Litoral, Santa Fe A3000, Argentina; imarcipr@fbcb.unl.edu.ar

**Keywords:** *Trypanosoma cruzi*, Chagas disease, live attenuated vaccine, saponin-based adjuvants, veterinary vaccine

## Abstract

Chagas disease, caused by the protozoan parasite *Trypanosoma cruzi*, remains a significant public health concern in Latin America, particularly impacting rural and marginalized populations. Among the various prophylactic strategies under investigation, vaccines based on live-attenuated parasites have demonstrated substantial potential to elicit protective immune responses. In the present study, we evaluated whether the inclusion of saponin-based adjuvants—ISPA and Quil-A—could enhance the immunogenicity and safety profile of a live-attenuated *T. cruzi* vaccine in a murine model. Immune responses were characterized at both early (post-vaccination) and long-term (one year) time points. All vaccinated groups exhibited robust protection against infection; however, mice receiving the adjuvants displayed a more tightly regulated immune profile, characterized by reduced tissue inflammation and more efficient parasite clearance. These results indicate that while the adjuvants did not markedly increase the magnitude of protection, they improved the quality and balance of the immune response and mitigated potential adverse effects. Collectively, these findings provide critical insights for the rational design of next-generation vaccines against Chagas disease, highlighting the role of adjuvants in optimizing long-term immunological outcomes in at-risk populations.

## 1. Introduction

Adjuvants are natural or artificial compounds used with the purpose of potentiating or modulating the immunogenic response to diverse types of antigens. These compounds can guide the type of immune response generated and increase the duration of protection as well as the number of individuals that respond to immunization [1]. For instance, subunit vaccines do not always elicit strong immune responses; therefore, the step of selection and incorporation of specific adjuvants to the formulations become a key factor for vaccine development [2]. Aluminum-based mineral salts (Alum) have been one of the most used adjuvants for years by several pharmaceutical companies. One of the properties characterizing this adjuvant is its capability of eliciting a good T_H_2 bias humoral response but a poor induction of cytotoxic T-cell immunity which is essential to defeat several intracellular pathogens [3]. This is the case for the protozoan parasite *Trypanosoma cruzi*, which has the ability to infect almost any mammalian cell, evade recognition by the host’s immune system and establish long-term persistence in the cellular cytoplasm, ultimately leading to myocardial complications in humans, known as Chagas disease [4,5]. Several immunoprophylactic vaccines—formulated with or without specific adjuvants—were evaluated over the years to prevent *T. cruzi* establishment in the host [6,7,8,9,10,11,12,13,14,15,16]. For instance, the TRASP chimeric protein from *T. cruzi* formulated with polyinosinic-polycytidylic acid stabilized with polylysine and carboxymethylcellulose (Poly-ICLC—Hiltonol) induces strong, long-lasting protection against parasite infection in mice and dogs, primarily via Cluster of Differentiation 8 (CD8^+^) T cells and interferon-gamma (IFN-γ) [17]. Heterologous prime-boost vaccination with Tc24 mRNA Lipid Nanoparticles and Tc24-C4 protein + glucopyranosyl A (GLA)-squalene emulsion elicits a broad, robust immune response in mice, including polyfunctional CD8^+^ T cells, balanced T_H_1/T_H_2/T_H_17 cytokines, and increased immunoglobulins profile (IgG, IgG1, and IgG2c), outperforming homologous protein, mRNA, or protein/mRNA regimens [18]. Nasal vaccination with trans-sialidase plus Cyclic di-adenosine monophosphate (c-di-AMP) induces strong mucosal and systemic immune responses, including T_H_1/T_H_2/T_H_17 cytokines as well and nasal IgA, and significantly reduces parasitemia, tissue parasite loads, and acute myocarditis in *T. cruzi*-infected mice [19]. Vaccination with a *T. cruzi*-specific protein family (TcTASV), delivered as recombinant proteins with aluminum hydroxide or via a recombinant baculovirus, elicited robust humoral and cellular immune responses against *T. cruzi*. Over 90% of vaccinated mice survived lethal challenges, showed reduced chronic tissue parasitism, and developed immunological memory that controlled infection reactivation and secondary challenges [20]. Collectively, these approaches not only highlight the versatility of adjuvant and vector design but also emphasize the central role of innate immunity and cytokine-driven modulation in shaping durable and protective responses against *T. cruzi*.

Saponins are natural products that possess—among other biological qualities—immunostimulant, antimicrobial, and antitumor activities [21]. Quil-A, which is a triterpenoid extracted from the tree *Quillaja saponaria* Molina, has a high cholesterol affinity and the ability to form immune stimulating complexes (ISCOMs). ISCOMs, in turn, can enhance antigen targeting, uptake and activity of antigen presenting cells—such as dendritic cells—by stimulating strong cellular and humoral responses, as well as differential antibody isotypes [22,23]. Quil-A has been extensively studied in various vaccine formulations targeting diseases relevant to humans [24,25,26]. Specifically, in the context of intracellular parasitic infections, it has been evaluated against malaria, toxoplasmosis, visceral leishmaniasis, and trypanosomiasis in mouse models [27,28,29]. In recent decades, cage-like structures composed of purified fractions of *Quillaja saponaria* extract, cholesterol, and phospholipids encapsulating selected antigens for vaccine formulation has garnered significant attention [23,30]. Bertona and cols. [31] generated a structure with these characteristics—called Immunostimulant Particle Adjuvant (ISPA for now on)—and compared it with other commercial adjuvants in vaccines approaches against *T. cruzi*. For this study, the authors used a member of the *T. cruzi* trans-sialidase superfamily (TS) previously assessed by their group and others as the selected immunogenic antigen. After experimental vaccination, data showed that the new adjuvant ISPA exceeded not only the performance of the other adjuvants in terms of type and speed of the immune response generated against the specific antigen, but also in the number of immunizations required to obtain high-quality antibody levels. After its development, this adjuvant was successfully used in several immunization and challenge studies against *T. cruzi* [15,32,33].

Through all this time, experimental immunization assays have provided a better understanding of trypanosomatids infection. Nowadays, it is well known that the immune response elicited by the natural infection of *T. cruzi* includes several humoral and cellular effector components that despite being the best immune actors capable of defeating infection, are not enough to completely eliminate the parasite [13]. Our group has been extensively studied a naturally attenuated strain of *T. cruzi* named TCC (TCC from *T. Cruzi* Culture). This strain has shown the ability to elicit a strong protective response, sufficient to efficiently control parasite replication, though not strong enough to induce sterile immunity in a mouse model [8,34,35,36,37,38]. Co-administration of additional immunostimulatory components such as a plasmid encoding murine IFN-γ or the parasite-derived recombinant P21 protein allowed us to modulate and enhance the immune response elicited by immunization with live TCC parasites [37,38]. The effects of next-generation adjuvants on live attenuated vaccine formulations have yet to be explored. Therefore, in the present work, we aimed to assess if the addition of two saponin-based adjuvants such as ISPA and Quil-A—which are similar in composition but different in terms of spatial conformation and modes of action—enhances the long-term protective efficacy of the TCC vaccine-induced response against *T. cruzi* without compromising its attenuated state. Additionally, to address safety concerns surrounding the use of live attenuated organisms in vaccines, we aimed to monitor TCC parasite presence in vaccinated animals. This served as an indicator of vaccine safety and allowed us to evaluate how adjuvants might influence parasite persistence or clearance from the host.

## 2. Materials and Methods

### 2.1. Institutional Ethical Statement

All animal protocols adhered to the National Institutes of Health (NIH) “Guide for the care and use of laboratory animals” and were approved by the School of Health Sciences and by the Ethical Committee of the Universidad Nacional de Salta, Salta, Argentina (Nº 311/18).

### 2.2. Animal Model

One-month-old C57BL/6 male mice were used throughout the immunization/challenge studies. Animals were housed in cages with up to 5 animals each and exposed to a 12 h light/dark cycle in a controlled temperature setting (25 °C) with free access to a standard laboratory chow diet and water. All the animals were bred at the Animal Facility of the Instituto de Patología Experimental “Dr. Miguel Angel Basombrío”, Universidad Nacional de Salta, Argentina.

### 2.3. Vaccine Formulation

*T. cruzi* epimastigotes from the naturally attenuated TCC strain were grown at 28 °C in liver infusion-tryptose medium (LIT) supplemented with 10% Fetal Bovine Serum (FBS) (Natocor, Cordoba, Argentina), 20 µg hemin, 100 IU of penicillin, and 100 µg/mL streptomycin. To obtain metacyclic trypomastigotes, axenic cultures containing epimastigote forms were allowed to age for a month. Complement-resistant forms were purified by incubating aged parasites with non-decomplemented FBS and human serum at a 1:1 ratio for 16 h at 37 °C. The parasites were then washed with phosphate-buffered saline (PBS) and quantified to determine the infectious dose for mouse inoculation. A total of 5 × 10^5^ TCC complement resistant forms were formulated either with 5 μg/dose of ISPA adjuvant [31] or 1 μg/dose of Quil-A adjuvant (InvivoGen, Toulouse, France).

### 2.4. Experimental Design for Short and Long-Term Prime/Boost and Challenge Infection

We conducted two independent experiments: one to determine the short-term efficacy of the vaccine and another to evaluate its long-term efficacy. For the short-term experiments, six groups of mice (*n* = 6 per group) were immunized with three doses administered subcutaneously at 15-day intervals. The groups were distributed as follows: G1: TCC + ISPA; G2: TCC + Quil-A; G3: TCC without adjuvants; G4: ISPA; G5: Quil-A; and G6: PBS (the latter four serving as control groups). Fifteen days after the final vaccination dose, three mice per group were euthanized by CO_2_ exposure. Blood was collected by cardiac puncture for subsequent serum isolation and IgG measurement (see Section 2.5); the spleen was collected for splenocyte isolation and cytokine measurement (see Section 2.6); the heart was taken for DNA extraction and parasite quantification by quantitative real-time PCR (qRT-PCR) (see Section 2.7); and skeletal muscle samples were collected for both histopathological analyses (see Section 2.8) and parasite load quantification by qRT-PCR (see Section 2.7). To evaluate the level of protection against *T. cruzi* infection following immunization, the remaining animals (*n* = 3) were challenged intraperitoneally with 100 bloodstream trypomastigotes/mouse of the highly virulent Tulahuen strain (see Section 2.9). For long-term vaccination efficacy, the same 6 experimental groups and immunization scheme was conducted (*n* = 6 per group). In this case, 12 months after the last immunization dose, three animals were euthanized, and the same samples were taken. At that time, the same virulent challenge was performed in the remaining animals (*n* = 3) (Figure 1A).

### 2.5. Specific IgG1, IgG2b and IgG2c Antibody Determination

Blood was collected by cardiac puncture and placed into 1.5 mL tubes, centrifuged at 3000 rpm for 10 min, after which the serum was separated and stored at −20 °C until use. Subtypes of immunoglobulin G antibodies against *T. cruzi* were measured by the enzyme-linked immunosorbent assay (ELISA) using total soluble proteins homogenate from the Tulahuen *T. cruzi* strain (HP-TUL). ELISA plates were coated with 1 μg of HP-TUL in 100 μL of Coating Buffer/ well and incubated overnight at 4 °C. Plates were then blocked with PBS containing 5% non-fat dry milk, followed by incubation with serum samples diluted 1:100 (100 μL/well) for 1 h at room temperature. After washing, plates were incubated for 1 h with biotin-conjugated goat anti-mouse IgG1, IgG2b or IgG2c antibodies (1:6000 dilution), followed by a 1 h incubation at 37 °C with streptavidin–horseradish peroxidase (HRP) conjugate. Color was developed with TMB Substrate Reagent Set (BD-Biosciences, USA) and monitored at 450 nm using a TECAN Infinite Pro microplate reader (Männedorf, Switzerland).

### 2.6. Splenocytes Cell Culture and Cytokines Measurements

Spleens were harvested from euthanized mice at either 2 weeks or 12 months post-immunization, placed on 15 mL tubes containing Roswell Park Memorial Institute medium (RPMI 1640) and gently macerated on a sterile mesh on ice. Cells were resuspended in a RPMI 1640 medium supplemented with L-glutamine (Biological Industries, Beit Haemek, Israel). Following centrifugation at 160× *g* at 4 °C for 10 min, cells were resuspended in lysis solution (0.17 M Tris, 0.16 M NH_4_Cl, pH 7.2) to remove erythrocytes. The remaining splenocytes were washed with RPMI and resuspended in RPMI supplemented with 10% FBS. The viability of cells was assessed by Trypan blue exclusion and cell number was determined by counting in a Neubauer chamber. Splenocytes (2 × 10^6^ cells/mL, in duplicate samples) were stimulated with 15 μg/mL of HP-TUL and incubated at 37 °C for 48 h at 5% CO_2_. Stimulation was also performed with 5 μg/mL of Concanavalin A (ConA, Sigma-Aldrich, Saint Louis, MO, USA) as a positive control. Cell culture medium was then collected after 48 h of stimulation and aliquots stored at −80 °C until their use for cytokine determinations. IL-10 and IFN-γ were determined by using optEIA ELISA kits (BD Biosciences, San José, CA, USA) according to the manufacturer’s specifications. Briefly, 96-well plates were coated overnight with each Capture Antibody in a 1:250 dilution. The following day, plates were blocked with PBS containing 10% fetal bovine serum (FBS) for 1 h at room temperature, then incubated with 100 μL per well of supernatant samples for 2 h at room temperature. A standard curve was incorporated at this point following the protocol indications. Detection Antibody (1:250 and 1:500 dilution, for IL-10 and IFN-γ, respectively) and streptavidin-horseradish peroxidase conjugate (1:250 dilution) were added and incubated simultaneously at room temperature for 1 h. For color development, TMB Substrate Reagent Set (BD-Biosciences, Franklin Lakes, NJ, USA) was used and incubated for 30 min until the reaction was stopped by the addition of 2 N H_2_SO_4_. Optical density was measured at 450 nm using a TECAN infinite f50 spectrophotometer and all measurements were interpolated to each standard curve.

### 2.7. Tissue Parasite Burden

We determined parasite burden in heart and skeletal muscle tissues using quantitative real-time PCR (qRT-PCR). Mice were sacrificed according to the specific experimental design: short-term (15 days after the last vaccine dose), long-term (12 months after the last vaccine dose), and post-challenge (25 days after virulent challenge). Hearts and skeletal muscle samples were collected, placed in 1.5 mL tubes, and stored at −80 °C until further use. Total DNA from tissues (50 mg) was isolated using DNA-Puriprep Highway nucleic acid T kit (InbioHighway, Tandil, Argentina), according to instructions provided by the manufacturer. qRT-PCR was performed on a QuantStudio5 thermal cycler (Applied Biosystems, Foster City, CA, USA) in a 20 μL reaction containing 40 ng of total DNA, 10 μL iTaq Universal SYBR Green Supermix (Bio-Rad, Hercules, CA, USA), and 1 µM rDNA SAT-*T. cruzi* specific oligonucleotides Fw (5′-GCAGTCGGCKGATCGTTTTCG-3′) and Rv (5′-TTCAGRGTTGTTTGGTGTCCAGTG-3′) [39]. PCR cycling conditions were as follows: initial denaturation at 95 °C for 10 min, followed by 40 cycles of denaturation at 95 °C for 15 s and annealing/extension at 63 °C for 30 s. Data were normalized to murine TNF-α amplification. For load quantification, a standard curve of total parasites was generated using serial dilutions. Data processing involved interpolating the Ct values of the samples with those obtained from the parasite standard curve [40]. In this way, the equivalent number of parasites per 40 ng of DNA was determined. To establish the positivity of a sample, the limit of detection (LoD) was calculated as follows: LoD = Mean Ct of the negative control—(3 × SD of the negative control), where SD equals Standard Deviation. Data was analyzed using the QuantStudio Design & Analysis 2.5.1 Software from Applied Biosystems (Applied Biosystems, Foster City, CA, USA).

### 2.8. Histopathology Evaluation

Skeletal muscle samples were taken 2 weeks (short-term) or 12 months (long-term) after the last vaccination dose and were fixed in 10% buffered formalin solution and dehydrated in a crescent concentration of ethanol solutions (70%, 80%, 95% and 100%), and embedded in paraffin blocks. Blocks were sectioned at a thickness of 3 μm and stained with Haematoxylin–Eosin (H&E). A double-blind microscopic evaluation of the entire tissue sections (*n* = 3 per group) was assessed on pre-coded slides. Mononuclear infiltrates were evaluated by light microscopy and scored as “+” to indicate the presence of small foci, or “−” when no infiltrate was detected. All muscle sections were also examined microscopically for the presence of intracellular parasites, with (−) indicating that no parasite nests were observed. Images were acquired using a Leica ICC50W camera (Leica, Wetzlar, Germany) attached to the light microscope.

### 2.9. T. cruzi Challenge, Parasitemia Assessment and Mortality Rate

To assess short and long-term protection, 15 days or 12 months after the final immunization, the remaining mice (*n* = 3) were challenged intraperitoneally with 100 bloodstream trypomastigotes from the infective Tulahuen strain (DTU VI), which was previously maintained by serial passages in C57BL/6 mice. Parasitemia was monitored by enumerating the number of parasites per 100 fields from 10 μL of fresh blood collected from the tail vein of anesthetized mice twice weekly under a light microscope (40× magnification). The daily survival rate was monitored until day 25, at which point individuals of the non-vaccinated group (PBS) died. Consequently, all remaining animals were sacrificed to prevent suffering and to address ethical considerations.

### 2.10. Safety of the Vaccine Formulations

Additionally, in a separate experiment, groups of 3 animals were vaccinated subcutaneously with three doses of the different formulations TCC, TCC + Quil-A or TCC + ISPA, at two-week intervals. For immunosuppressive regimens, Cyclophosphamide (Cy) (Endoxán-Cyclophosphamide, Labinca, Buenos Aires, Argentina) has been diluted in sterile water. Two weeks after the last immunization dose, immunized mice were injected intraperitoneally with 350 mg/Kg/mouse of Cy, every other day for 10 days. Ten days after the last Cy dose, mice were sacrificed, and target organs were collected for *T. cruzi* detection by qRT-PCR analysis as described above (Figure 1B).

### 2.11. Statistical Analysis

All analyses were performed using the GraphPad Instat 8.0 software (GraphPad, San Diego, CA, USA). Data distribution was assessed using the Shapiro–Wilk normality test. Statistical significance was established as *p* < 0.05. All datasets were subjected to a one-way analysis of variance (ANOVA) followed by Tukey’s post hoc honestly significant difference test for inter-group comparisons. Survival curves were evaluated using the Mantel–Cox (Log-rank) test. Data is expressed as the mean ± standard error of the mean (SEM) from at least three independent experiments unless otherwise stated.

## 3. Results

### 3.1. Immunization with Attenuated Trypanosoma cruzi TCC Parasites Combined with Saponin-Based Adjuvants Effectively Reduce Tissue Inflammation and Long-Term Parasite Persistence

We initially focused on assessing parasite presence by quantifying *T. cruzi* satellite DNA using qRT-PCR, as well as evaluating potential tissue damage and persistence caused by the TCC immunizing parasites during the vaccination period as depicted in Figure 1A. Two weeks after vaccination, immunizing parasites were detected in all TCC-inoculated animals (with and without adjuvants) at least in one of the analyzed mice. However, twelve months later, the parasite levels in the skeletal muscle and heart of vaccinated animals remained negligible and even Ct values were not significantly different from those in non-vaccinated animals, indicating effective parasite clearance over the evaluation period (Table 1). These results suggest that immunization with TCC-attenuated parasites—whether administered with or without adjuvants—may represent a safe vaccination strategy in immunocompetent mice, as the inoculated parasites remained undetectable even one year post-vaccination.

After assessing parasite burden via qRT-PCR, we performed a histological examination of skeletal muscle tissue sections to evaluate tissue damage and presence of parasite nests in immunocompetent mice immunized with live attenuated parasites, with or without the presence of adjuvants. In the short-term analysis, skeletal muscle sections from TCC, TCC + ISPA, and TCC + Quil-A immunized animals showed minimal residual inflammatory infiltrates, primarily consisting of mononuclear cells infiltrates, as illustrated in Figure 2A (upper panel) and detailed in Figure 2B. One-year post-vaccination, only skeletal muscle from TCC-immunized animals exhibited mild inflammatory infiltrates, with mononuclear cells concentrated in small foci between muscle fibers (Figure 2A, lower panel, and Figure 2B). Notably, the addition of ISPA or Quil-A had a lasting positive impact, significantly reducing infiltrates, with no amastigotes nests detected in any of the tissue sections examined under microscopy.

To assess the safety of TCC vaccination in immunocompromised mice, immunized animals were treated with cyclophosphamide (Figure 1B). Following immunosuppression, only one mouse in the TCC group exceeded the detection limit. Remarkably, combining ISPA or Quil-A with TCC-attenuated parasites yielded undetectable parasite DNA in all analyzed tissues of immunocompromised mice (Table 2).

### 3.2. ISPA and Quil-A Adjuvants Beneficially Modify the Humoral and Cellular Immune Profile Induced by Immunization with Attenuated T. cruzi TCC Parasites

To determine whether the co-administration of ISPA or Quil-A adjuvants have modulatory effects in the humoral response elicited by immunization with TCC attenuated parasites, we analyzed the levels of *T. cruzi*-specific IgG subtypes after prime and boost immunization schemes. Soon after vaccination, IgG1 levels were significantly higher in the TCC-adjuvanted groups compared to TCC control group (Figure 3A). In contrast, IgG2b levels were modest while IgG2c levels were nearly absent in these groups (Figure 3B,C). At later time points (long-term vaccination), although overall antibody levels declined, the TCC + ISPA group maintained a predominance of IgG1 and IgG2b, while IgG2c levels remained at lower concentrations. In the case of the TCC + Quil-A group, low levels of IgG1 and IgG2b were observed, but a slight increase in IgG2c was detected (Figure 3D–F). As a general overview, in the short-term, animals receiving only TCC displayed an increased IgG2b/IgG1 and IgG2c/IgG1 ratio compared to the adjuvanted groups, suggesting that the T_H_1-biased response elicited by TCC was weakened by ISPA or Quil-A (Figure 3G). However, a year post-vaccination, all groups that received the adjuvants shifted from a T_H_2-like profile to a T_H_1-biased response over time (Figure 3H).

Two weeks or twelve months after the last immunization dose, mice were euthanized (*n* = 3) to analyze ex vivo IL-10 and IFN-γ cytokine production by stimulated splenocytes by ELISA. Regarding IL-10 secretion shortly after vaccination, the influence of ISPA and Quil-A on the TCC response (compared to the TCC control group) is evident, as IL-10 levels drop in mice vaccinated with attenuated parasites plus adjuvants (Figure 4A). This trend is not sustained over time; after twelve months, IL-10 levels increase in the TCC + ISPA group while decreasing in mice immunized only with TCC. Animals treated with TCC + Quil-A maintained consistently low values, similar to those observed in the short-term analysis (Figure 4C). In terms of IFN-γ production shortly after vaccination, no significant levels were detected across the groups, and the addition of adjuvants did not lead to any notable impact on IFN-γ response (Figure 4B). Nevertheless, one year after immunization, splenocytes from all experimental groups displayed reactivity to *T. cruzi* protein homogenate stimulation, with increased IFN-γ levels, showing no differences compared to the TCC control group (Figure 4D).

### 3.3. Immunization with TCC and ISPA or Quil-A Adjuvants Reduces Spread and Tissue Colonization of Virulent Parasites in Animals

Mice primed with TCC, TCC + ISPA, and TCC + Quil-A and challenged shortly after vaccination (two weeks post-vaccination) demonstrated 100% survival throughout the follow-up period (Figure 5A), with scarcely detectable levels of circulating parasites in peripheral blood (Figure 5B). The follow-up continued until the control group exhibited significant distress or began to die, at which point euthanasia was applied to minimize suffering. Remarkably, even one year after the vaccine administration, all mice inoculated with TCC—with or without adjuvants—and subsequently exposed to Tulahuen infection survived and effectively controlled parasite replication (Figure 5D,E). To evaluate the impact of different vaccination approaches on reducing the overall parasite load, we calculated the area under the parasitemia concentration-time curve (AUC). All TCC-vaccinated groups exhibited statistically significant differences compared to non-vaccinated control groups, demonstrating a reduction in parasite load exceeding 98%. In particular, no statistically significant differences were observed among the TCC-vaccinated groups either in the short or long-term immunization and challenge models (Figure 5C,F).

Further, we conducted qRT-PCR analysis on heart and skeletal muscle samples from vaccinated and challenged animals. In the short term, we observed that parasite load was minimal in heart samples and was primarily detected in skeletal muscle of non-vaccinated animals (Figure 6A,B). Notably, no parasites were detected in any of the TCC-vaccinated animals, with or without adjuvants. In the long term, the parasite burden in both organs was significantly lower in all TCC-vaccinated animals than in non-immunized animals (*** *p* < 0.001). In fact, *T. cruzi* DNA levels in TCC-vaccinated groups were only 1% of the load detected in the hearts and in the muscles than in the non-vaccinated groups. When comparing samples from the TCC and TCC-adjuvanted groups, no differences were observed in the analyzed organs (Figure 6C,D).

## 4. Discussion

In this study, we aim to investigate how the addition of saponin-based adjuvants, such as ISPA and Quil-A, affects the protective immune response elicited by immunization with attenuated parasites of the TCC strain of *Trypanosoma cruzi* and their capacity to safeguard mice against virulent infection. Our study provided an overall assessment, considering both immediate changes in immunogenicity and vaccine efficacy following immunization, as well as the long-term protective efficacy against virulent infection. This dual focus is crucial for advancing vaccine development strategies for Chagas disease whether for human or veterinary applications [41]. Additionally, we conducted an in-depth evaluation using a highly sensitive immunosuppression model to investigate whether the immunizing parasites persisted in immunocompromised hosts and to elucidate the role of adjuvants during this critical phase of the study.

Achieving complete clearance of the parasite from the host has long been the goal of researchers developing various vaccine formulations. However, despite decades of effort, progress in this area remains slow and far from ready for implementation [16]. The primary challenges lie in identifying the optimal combination of antigens and adjuvants, as well as overcoming the limitations of current vaccine delivery systems [16]. Moreover, the parasite’s remarkable ability to evade the immune system and persist in a dormant state has further hindered efforts to develop an effective vaccine [42]. Our findings indicate that effective parasite clearance or control requires both a broad range of antigens and full parasite-host interactions to stimulate a robust immune response [8,10,34,35,36,37,38,43]. Supporting this, other research shows that dead or fixed parasites fail to provide protection in experimental infections [44]. Therefore, live vaccines offer important benefits but come with challenges. Ensuring long-term safety and protective efficacy is critical for success, yet this has not been adequately addressed in the development of a Chagas disease vaccine. While live-attenuated vaccines are designed not to cause disease in immunocompetent hosts, they pose a risk to immunocompromised individuals due to the potential for re-emergence or reversion to a virulent form. To address safety concerns in our study, we performed qRT-PCR and histological analyses on target organs from TCC-vaccinated animals. After initial vaccination, parasites may still be detectable, likely because the immune system requires additional time to mount a fully effective response. Recent studies indicate that parasites often establish a brief period of proliferation before the immune response is fully mobilized to control the infection [45]. Even with primed immune cells at the infection site, recognizing and eliminating parasites often demands at least one cycle of host cell infection and parasite expansion, allowing the infection a chance to establish and potentially spread. However, one year post-vaccination, immunizing parasites were no longer detectable in immunocompetent hosts, and no significant inflammatory responses were observed. This highlights the efficacy of TCC formulations combined with saponin-based adjuvants, which produced reduced inflammation in skeletal muscle tissue compared to TCC-only vaccinations. A highly novel finding in this study was the absence of parasites in the tissues of immunosuppressed animals immunized with TCC combined with the different adjuvants, in contrast to those immunized with TCC alone. These results further underscore the potential of adjuvants such as ISPA and Quil-A to enhance the safety profile of live attenuated vaccines.

The role of cell-mediated immunity in protection against *T. cruzi* infection is well established. Previous studies have shown that the robust T cell response elicited during natural *T. cruzi* infection provide significant, though non-sterilizing, protective immunity against reinfection [5,46]. These T cells secrete key cytokines like IFN-γ and TNF-α, which are critical for controlling and eliminating intracellular parasites [47]. Although humoral immunity was once considered less relevant, antibodies, particularly IgG, play an essential role by targeting extracellular parasite forms, including circulating trypomastigotes, thereby limiting parasitemia and preventing dissemination [48]. Thus, both cell-mediated and humoral responses are indispensable components of the coordinated defense against *T. cruzi*. Therefore, selecting adjuvants that enhance and modulate immune response quality is essential for increasing antigen efficacy and durability. While natural immunity from infection remains the gold standard for immune protection, live attenuated vaccines, such as TCC, could significantly benefit from adjuvants that boost efficacy and facilitate pathogen clearance.

Saponin-based adjuvants such as Quil-A enhance vaccine efficacy by stimulating type I and II interferon secretion, improving antigen presentation, and promoting cytotoxic T lymphocyte responses [41]. Similarly, immunostimulatory complexes (ISCOMs) like ISPA amplify immunity by driving IgM, IgA, and IgG production, stimulating T cell proliferation, and fostering a balanced T_H_1/T_H_2 response [16,25]. In our study, vaccination with attenuated TCC parasites alone elicited only a modest T_H_1-like response, as indicated by the IgG2/IgG1 ratio. The addition of ISPA or Quil-A initially shifted the profile toward T_H_2 dominance, an effect more evident with ISPA. By one year post-vaccination, however, both adjuvant groups displayed a balanced T_H_1/T_H_2 profile, likely reflecting the decline in adjuvant bioavailability over time. While Quil-A has a short half-life dependent on formulation and administration route, ISPA’s particulate nature may provide more sustained immune stimulation, although its pharmacokinetics are less well characterized. These findings are consistent with previous reports describing Quil-A’s ability to promote antibody generation and establish balanced T_H_1/T_H_2 immunity [42,43,44]. In this context, the measurement of IgG1 and IgG2 subclasses provides a useful readout of humoral polarization during TCC vaccination. In *T. cruzi* models, IgG1 and IgG2 are established markers of T_H_2- and T_H_1-biased responses, respectively [49,50,51], and both ISPA and Quil-A are known to modulate their balance. Mechanistically, ISPA’s nanoparticulate structure enhances antigen presentation by dendritic cells and induces both CD4^+^ and CD8^+^ T cell responses. This mixed T_H_1/T_H_2 activation supports the generation of IgG1 as well as IgG2b/IgG2c, promoting antigen persistence and long-term antibody maintenance. Quil-A, though acting through different pathways, also induces IgG1, IgG2b, and IgG2c via cytokine-driven mechanisms. Thus, despite their distinct modes of action, both adjuvants create an immunological milieu that favors the induction and persistence of multiple IgG subclasses, ultimately supporting a more robust and durable protective immune response. In this model, the relative distribution of IgG1 and IgG2 therefore reflects not only the adjuvant effect but also serves as a functional marker of sustained vaccine-induced immunity.

In line with the points mentioned above, an effective immune response against *T. cruzi* requires the production of specific cytokines [12]. Both innate and adaptive immune cells secrete cytokines that play crucial roles in signaling, recruiting other cells, and ultimately clearing the parasite [5]. During the initial weeks of infection, macrophage activation through cytokines such as IFN-γ is primarily responsible for controlling parasitemia while IL-10 plays a critical role in preventing excessive tissue damage caused by an overactive immune response [52]. However, during *T. cruzi* infection, its role is complex and can be both protective and detrimental depending on the stage of infection [53]. ISPA and Quil-A exert their immunostimulatory activity largely through the modulation of cytokine responses. According to previous evidence, both adjuvants—ISPA and Quil-A—support IL-10 production and secretion of IFN-γ, which would contribute to a balanced and regulated immune response, crucial for both effective immunity and the prevention of immune-related damage in infections like *T. cruzi*. In our experiment, in the short term, co-administration of either ISPA or Quil-A with the attenuated TCC parasites reduced IL-10 production without significantly affecting IFN-γ levels, suggesting that these adjuvants initially dampen regulatory pathways without amplifying T_H_1 responses. By contrast, one year after immunization, only the TCC + ISPA group showed restored IL-10 levels, pointing to a delayed regulatory feedback specific to this formulation. Such recovery of IL-10 may help sustain long-term antibody responses while limiting immunopathology, highlighting the differential and time-dependent roles of saponin-based adjuvants in shaping cytokine-mediated immunity. This finding is consistent with reports describing the T_H_1-skewing properties of saponin adjuvants [48,49]. Importantly, our long-term data suggest that persistent IL-10 production in TCC- and TCC + ISPA-vaccinated animals reflects a balanced immune state: strong inflammatory responses are needed initially to control attenuated parasites, but regulatory mechanisms must subsequently take over to prevent tissue damage, as supported by our histopathological observations. In this context, IL-10 acts as a double-edged sword protecting the host from immune-mediated injury while also potentially favoring parasite persistence and contributing to the chronicity of Chagas disease. Additional studies are required to clarify whether the long-term increase in IL-10 ultimately benefits or compromises disease outcome. In the present study, we limited our analysis to IFN-γ and IL-10, based on prior evidence indicating that these cytokines represent key markers of the immune response to TCC infection and may serve as sensitive indicators of the modulatory effects of adjuvants over time [38,54]. We recognize, however, that the evaluation of only two cytokines constitutes a limitation, as it does not provide a complete picture of the immune response. A more comprehensive assessment including additional cytokines such as TNF-α, IL-4, and IL-12 would be necessary to better define the T_H_1/T_H_2 balance and to strengthen the interpretation of the humoral and cellular immune profiles elicited by vaccination.

Although differences in cytokine and antibody production levels were observed, the analysis of parasitic load in the heart and skeletal muscle of immunized animals did not reveal any statistically significant differences to determine a clearly superior formulation. Both live parasites alone and those combined with ISPA or Quil-A provided comparable levels of protection. However, the long-term absence of detectable immunizing parasites and the lack of tissue damage—likely attributed to the elevated IL-10 levels induced by the immune response—suggest that the TCC + ISPA formulation may represent the most effective option. This formulation not only ensures protection but also mitigates potential risks associated with chronic inflammation and tissue pathology, highlighting its promise for future vaccine development strategies. Nevertheless, additional parameters indicative of protection, such as IgG expansion, histopathological evaluation, or clinical assessments including electrocardiography, would be valuable and necessary to achieve a more comprehensive determination of the efficacy of our vaccine formulations.

In summary, we present a vaccine formulation consisting of live attenuated parasites combined with saponin-based adjuvants, providing sustained immunity and rapid recall responses against *T. cruzi* infections in mice. Advances in biosafety techniques have alleviated concerns that this prolonged immunity is due to parasite persistence, underscoring the safety of this approach. The attenuated *T. cruzi* TCC strain has been a cornerstone of our research, consistently delivering exceptional results in experimental vaccination models. Its demonstrated ability to elicit robust immune responses and confer protection in preclinical studies highlights its potential as a foundation for future vaccine development [10,34,35,36,37,38,43]. By pairing the attenuated TCC strain with saponin-based adjuvants, such as ISPA and Quil-A, we aimed to enhance its immunological properties while improving vaccine safety. Our findings reveal that combining ISPA or Quil-A with the TCC strain provides a level of protective immunity comparable to the TCC vaccine alone, maintaining strong defenses against the parasite. Despite the challenges posed by publication pressures, long-term studies remain critical for advancing vaccine development. Evaluating the durability and safety of vaccination strategies over extended periods is essential to achieving meaningful progress toward an effective vaccine. While a live attenuated vaccine for human use is not yet within reach, immediate efforts should prioritize the development of a blocking veterinary vaccine [55,56]. Such a vaccine could significantly limit parasite transmission by triatomine vectors, playing a pivotal role in controlling *T. cruzi* infections at the household level in endemic regions.

## 5. Conclusions

The results of this work highlight the promise of the attenuated *T. cruzi* TCC strain as a key component in future vaccine strategies. Its combination with saponin-based adjuvants, such as ISPA and Quil-A, preserves strong immune protection while contributing to improved safety profiles. Current biosafety advancements help address past concerns about the persistence of live parasites, supporting the feasibility of this approach. Although a vaccine for human use may still be years away, a more immediate and achievable goal is the development of a veterinary vaccine. Such an intervention could significantly reduce domestic transmission by insect vectors in endemic regions, serving as a valuable tool in the broader effort to control Chagas disease. Long-term studies, though often constrained by publication demands, remain essential for evaluating the durability and safety of vaccine candidates and for moving the field closer to effective, real-world solutions.

## Figures and Tables

**Figure 1 biology-14-01298-f001:**
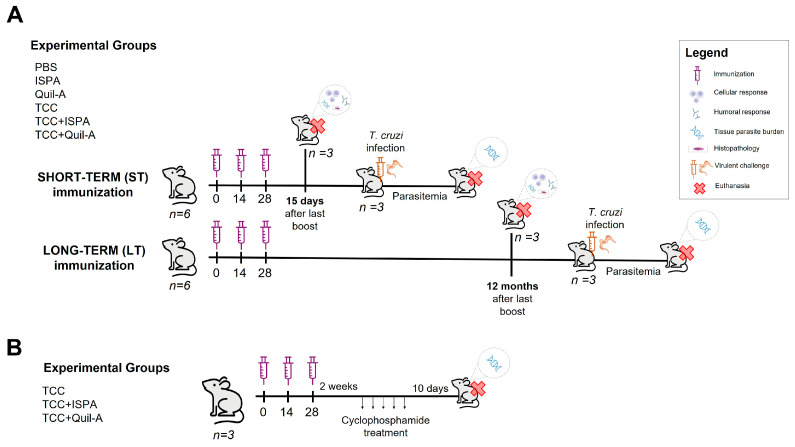
Schematic overview of in vivo evaluation of TCC plus adjuvant immunization efficacy and safety. Animals were primed and boosted with formula containing 5 × 10^5^ live complement resistant forms of the *T. cruzi* TCC attenuated strain supplemented with either ISPA or Quil-A adjuvants for short and long-term schemes (**A**). Details of the immunization scheme and further immunosuppression regimen to evaluate safety (**B**). Red cross means animals were euthanized at that time point and different samples were taken for diverse analyses, detailed in legend.

**Figure 2 biology-14-01298-f002:**
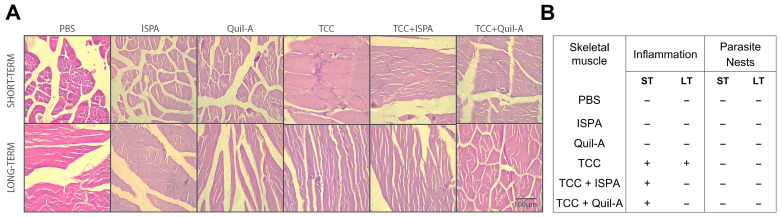
Parasite persistence and tissue damage in TCC plus adjuvants immunized mice. Representative hematoxylin and eosin-stained skeletal muscle sections from all groups, imaged at 40× magnification. Short-term (upper panel) and long-term (lower panel) schemes (**A**). Muscle sections from vaccinated animals were examined for cellular infiltration as indicator of damage, and intracellular parasite nests as indicator of parasite persistence. Infiltration was scored as “+” for small foci and “−” when absent. For parasite nests, “−” indicated their absence (**B**).

**Figure 3 biology-14-01298-f003:**
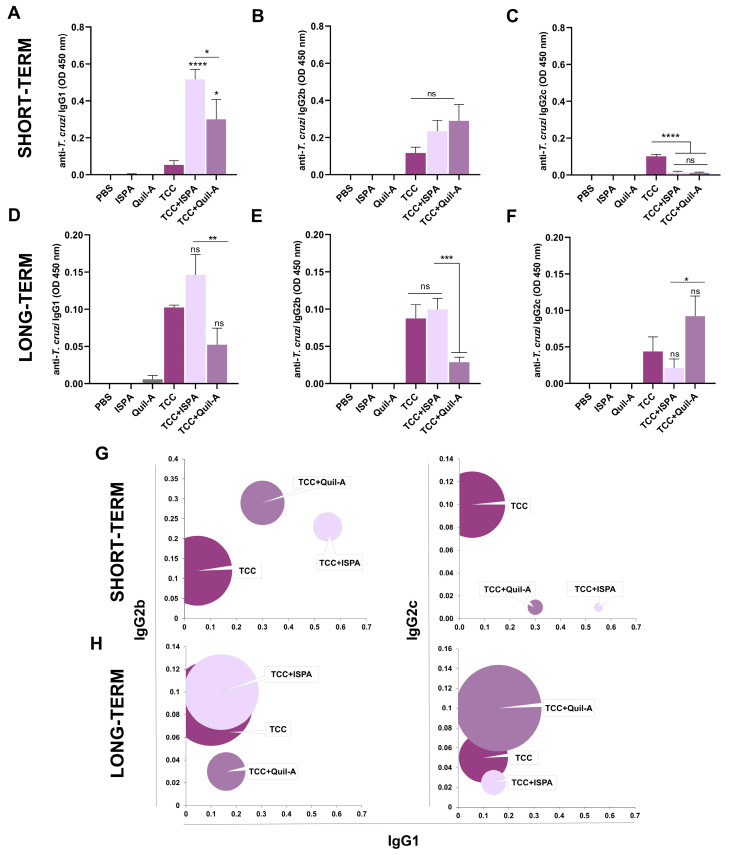
*T. cruzi*-specific humoral response following administration of ISPA and Quil-A adjuvants in combination with TCC parasites. Serum samples were collected two weeks or twelve months after the last immunization dose for measurement of specific anti-*T. cruzi* IgG1 (**A**,**D**), IgG2b (**B**,**E**), and IgG2c (**C**,**F**) antibodies. IgG2b/IgG1 and IgG2c/IgG1 ratios in both short-term (**G**) and long-term vaccination periods (**H**). Values are expressed as means with standard errors of the mean (SEM). Asterisks denote statistical significance compared to the TCC control group (**** *p*< 0.0001, *** *p*< 0.001, ** *p* < 0.01, * *p* < 0.05, ns, not significantly different) or between groups when specified.

**Figure 4 biology-14-01298-f004:**
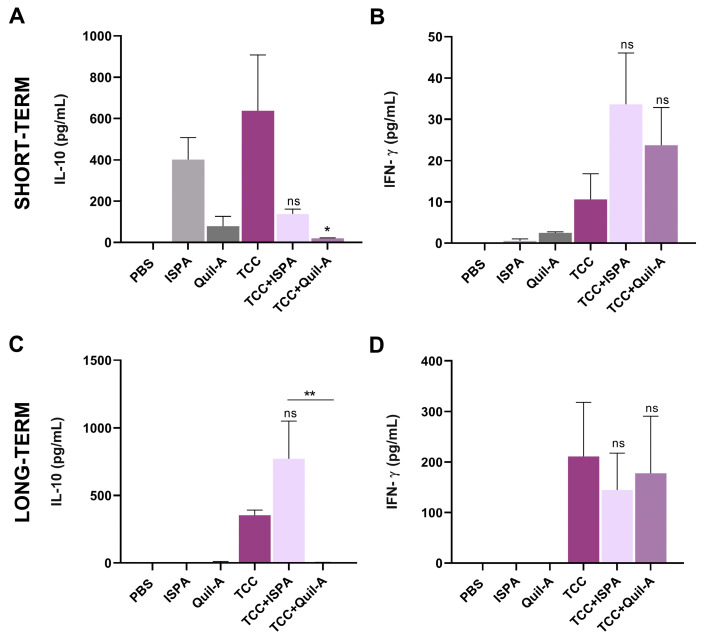
IL-10 and IFN-γ levels in mice immunized with TCC attenuated parasites co-administered with ISPA and Quil-A adjuvants. Levels of IL-10 and IFN-γ in splenocyte culture supernatants were measured by ELISA from spleens harvested two weeks after immunization (**A**,**B**) and 12 months after the last immunization dose (**C**,**D**). Values are expressed as means with SEM. Asterisks denote statistical significance compared to the TCC control group (** *p* < 0.01, * *p* < 0.05, ns, not significantly different) or between groups when specified.

**Figure 5 biology-14-01298-f005:**
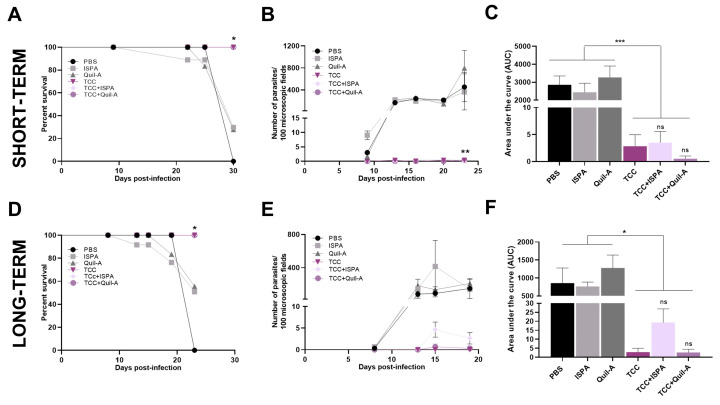
Mortality rate and parasite load in animals primed with TCC, TCC + ISPA, and TCC + Quil-A following *T. cruzi* virulent challenge. Survival rates were monitored daily during the acute phase of infection in mice challenged 2 weeks (**A**) or 12 months (**D**) after the last vaccination dose. Data analysis was performed using Log Rank Test with * *p* ≤ 0.05. Parasitemia curve was obtained during the acute phase of the infection in the short-term (**B**) and long-term (**E**) vaccination scheme. For statistical analysis 2-way ANOVA plus Tukey’s test were performed. Concentration-time curve (AUC) of the parasitemia curve short-term (**C**) and long-term (**F**) vaccination scheme. Values were tested by One-way ANOVA plus Tukey’s post-test. Data are expressed as the mean ± SEM of two independent experiments. Asterisks denote statistical significance among groups (* *p* < 0.05, ** *p* < 0.01, *** *p* < 0.001; ns, not significantly differences among TCC groups).

**Figure 6 biology-14-01298-f006:**
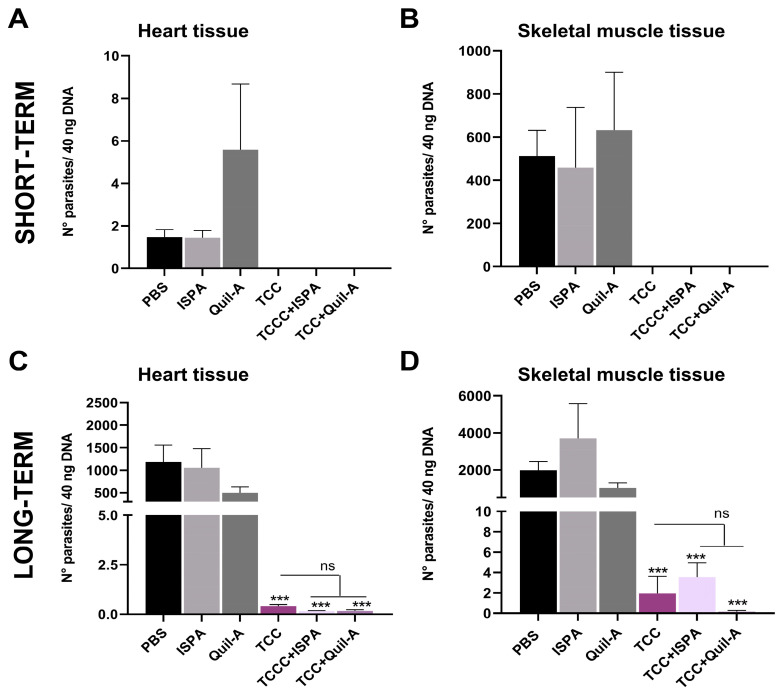
Quantification of parasite numbers using quantitative real-time PCR in animals primed with TCC, TCC + ISPA, and TCC + Quil-A after a virulent challenge. Number of parasites detected by *T. cruzi* satellite DNA amplification in heart and skeletal muscle from mice challenged 2 weeks ((**A**) and (**B**), respectively) and 12 months after vaccination ((**C**) and (**D**), respectively). Data are expressed as the mean ± SEM, asterisks denote statistical significance compared to the PBS non-vaccinated control group (*** *p* < 0.001) or between groups when specified; ns, not significantly differences among TCC groups.

**Table 1 biology-14-01298-t001:** Presence of TCC parasites in immunocompetent immunized animals. Number of qRT-PCR–positive mice out of the total animal analyzed (*n* = 3 per group).

		PBS	ISPA	Quil-A	TCC	TCC + ISPA	TCC + Quil-A
**Short-term**	Heart	0/3	0/3	0/3	3/3	1/3	2/3
	Muscle	0/3	0/3	0/3	1/3	2/3	1/3
**Long-term**	Heart	0/3	0/3	0/3	0/3	0/3	0/3
	Muscle	0/3	0/3	0/3	0/3	0/3	0/3

**Table 2 biology-14-01298-t002:** Detection of TCC parasites in immunized mice following cyclophosphamide-induced immunosuppression. Data represent the number of qRT-PCR–positive animals out of the total analyzed (*n* = 3 per group).

		TCC	TCC + ISPA	TCC + Quil-A
**Immunosuppression**	Heart	1/3	0/3	0/3
	Muscle	1/3	0/3	0/3

## Data Availability

All data is available upon reasonable request to the authors.

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
