# Peer review of "Induction of Sustained Immunity Following Vaccination with Live Attenuated Trypanosoma cruzi Parasites Combined with Saponin-Based Adjuvants"

_biology, 2025, doi:10.3390/biology14091298_

Round 1
Reviewer 1 Report
Comments and Suggestions for Authors
A vaccine against Chagas disease is needed for both human and veterinary use. This study does a good job of initiating the prophylactic evaluation of a live-attenuated vaccine option against T. cruzi.
Below, you can find a few comments and suggestions.
For section 2.6. histopathology evaluation, please specify when the long-term study tissues were collected; only short-term collection of tissues was specified. I could determine the collection for long-term tissues when I read section 2.4.
I disagree with the statement that this is a comprehensive study. Also, the discussion does not list the limitations of the study. Here are a few:
1) n=3 per group is a small sample size.
2) two antibodies and two cytokines measured,
Measuring two cytokines and two antibodies provides very limited insight into Th1 and Th2 immune responses, and it is therefore difficult to give a definitive determination of the overall Th1/Th2 balance or a comprehensive picture of the immune response.
4) There was no evaluation of histopathology, antibodies or cytokines of short-term and long-term mice after Tulahuen challenge in this study
Measuring percent survival, circulating parasites and quantification of tissue parasites is not sufficient to assess the efficacy of the prophylactic vaccine in mice. It’s a good start and they are valuable indicators of vaccine performance, but a more comprehensive approach is needed.
Author Response
A vaccine against Chagas disease is needed for both human and veterinary use. This study does a good job of initiating the prophylactic evaluation of a live-attenuated vaccine option against T. cruzi. Below, you can find a few comments and suggestions.
For section 2.6. histopathology evaluation, please specify when the long-term study tissues were collected; only short-term collection of tissues was specified. I could determine the collection for long-term tissues when I read section 2.4.
Thank you for your positive assessment of our work and for your constructive comments. We agree that a vaccine against Chagas disease is urgently needed, and we appreciate your recognition of our efforts to evaluate a live-attenuated vaccine candidate.
We also thank the reviewer for pointing out the absence of this information in the indicated section. This has now been incorporated into the manuscript with track changes (section 2.6).
I disagree with the statement that this is a comprehensive study. Also, the discussion does not list the limitations of the study. Here are a few:
1) n=3 per group is a small sample size.
We agree with the reviewer that "comprehensive" is possibly not the most accurate description for our study. We acknowledge that the number of immunological markers used here is limited. However, we believe this study represents a valuable preliminary evaluation of a live-attenuated vaccine for long-term protection against T. cruzi
Regarding sample size, this could also be a study limitation. However this small n used is in accordance with the 3Rs principle (Replacement, Reduction, and Refinement), and following the requirements of our Institutional Ethics Committee. This number was considered sufficient to generate preliminary yet consistent data while minimizing animal use and suffering. Taking this in consideration, our experimental design was guided by both ethical considerations and institutional regulations, ensuring compliance with the principles of responsible animal research.
2) two antibodies and two cytokines measured. Measuring two cytokines and two antibodies provides very limited insight into Th1 and Th2 immune responses, and it is therefore difficult to give a definitive determination of the overall Th1/Th2 balance or a comprehensive picture of the immune response.
We acknowledge the reviewer’s concern that assessing only two antibody isotypes and two cytokines provides limited insight into the overall Th1/Th2 balance and does not allow for a comprehensive characterization of the immune response. We agree that, ideally, a broader panel would provide more detailed information. However, in the context of evaluating long-term memory responses during an established T. cruzi infection, IgG1 and IgG2 are considered the most informative and widely accepted markers to assess Th1- versus Th2-associated humoral immunity. Indeed, this approach is routinely used in T. cruzi immunological studies to evaluate vaccine-induced or infection-driven immune profiles (doi: 10.3389/fimmu.2019.01456; https://doi.org/10.1016/j.vaccine.2018.11.041; doi: 10.1371/journal.pntd.0003625). Nevertheless, we have expanded our dataset by including measurements of IgG2b, which may further enrich the interpretation of the humoral response.
Regarding cytokines, even when two molecules assessment could be limited, based on our previous studies in mice infected with the attenuated TCC strain, we have observed that IFN-γ production was not evident at early time points but emerged later during the course of infection, highlighting its relevance as a time-dependent marker (doi: 10.1590/0074-02760180571; doi: 10.1186/s12879-017-2834-6; doi: 10.1007/s004360050524). Similarly, IL-10 was identified as a key cytokine contributing to immune regulation in this model. Taken together, we considered IFN-γ and IL-10 as essential indicators of the modulatory effects of the adjuvants on the immune response.
In any case, we have modified the Discussion to explicitly acknowledge these limitations, as suggested by the reviewer.
3) There was no evaluation of histopathology, antibodies or cytokines of short-term and long-term mice after Tulahuen challenge in this study. Measuring percent survival, circulating parasites and quantification of tissue parasites is not sufficient to assess the efficacy of the prophylactic vaccine in mice. It’s a good start and they are valuable indicators of vaccine performance, but a more comprehensive approach is needed.
We fully agree with the reviewer that evaluating histopathology, antibody responses, and cytokine profiles in both the short and long term after Tulahuen challenge would provide a more comprehensive assessment of vaccine efficacy. We greatly appreciate this valuable comment. However, we consider that parasite burden is the primary and most critical parameter to assess in a first instance, as a reduction in parasitemia is a prerequisite before further evaluating complementary parameters such as histopathology or antibody expansion. Without a sustained reduction in parasite load, measuring these additional outcomes would have limited meaning. Nonetheless, we recognize the importance of the reviewer’s suggestion and agree that incorporating these additional analyses would be highly valuable to complement and strengthen our findings. We will certainly take this into account for future experiments. In addition, the discussion section has been revised to explicitly include this limitation of the present study. Finally, we believe that transparently communicating scientific results—including limitations—is key for promoting the collective advancement of science. Under this view, we believe that incorporating our study limitations is an improvement.
Reviewer 2 Report
Comments and Suggestions for Authors
As per the attached file

Author Response
We appreciate the reviewer's thorough evaluation and are grateful for their insightful comments. All grammatical and wording suggestions (highlighted in the pdf file) have been addressed as well as figure descriptions and results.
Line 41. Check the time of immune response assessment in short term study
We thank the reviewer for noticing this writing error. It has now been appropriately corrected.
Line 175. Why was only skeletal muscle examined? What about the heart?
We selected skeletal muscle as the primary tissue for the search of attenuated parasites following the vaccination regimen, given that it typically harbors the highest parasite burden and displays the most pronounced cellular infiltration and intergroup differences. Furthermore, T. cruzi persistence has been well-documented in skeletal and intestinal muscle cells, with occasional involvement of cardiac tissue, emphasizing the relevance of investigating this site. In light of our objective to identify attenuated parasites, these preliminary analyses were therefore focused on this tissue, where the probability of detection is expected to be highest
Line 183. The statement is not clear and it has to be modified
Modified as suggested.
Reviewer 3 Report
Comments and Suggestions for Authors
This manuscript, entitled “Induction of long-lasting sterilizing immunity following vaccination with live attenuated Trypanosoma cruzi parasites combined with saponin-based adjuvants”, presents a comprehensive study evaluating the use of saponin-based adjuvants (ISPA and Quil-A) in combination with live attenuated Trypanosoma cruzi (TCC) parasites to enhance vaccine safety and immune quality in a murine model. While live attenuated vaccines for T. cruzi are promising, they raise important safety concerns. The authors’ focus on saponin-based adjuvants to mitigate inflammation and promote a balanced immune response is innovative. However, the manuscript would benefit from the following clarifications and revisions:
- Several sections suggest sterilizing or superior protection; however, the Results/Discussion indicate comparable protection between TCC and adjuvanted groups, with no significant differences in parasite loads post-challenge. The authors should reconcile these points and temper claims of “long-lasting protective immunity.”
- The methods present two different challenge doses: 100 trypomastigotes/mouse in Section 2.4 versus 1000 in Section 2.11. The correct challenge dose and timing should be clarified and reported consistently.
- The antibody isotype and cytokine data reveal nuanced differences in immune modulation by ISPA versus Quil-A. The discussion would be strengthened by explicitly linking these observations to known adjuvant mechanisms (e.g., ISCOM-mediated antigen presentation for ISPA, or Quil-A’s influence on TH1/TH2 polarization).
Author Response
This manuscript, entitled “Induction of long-lasting sterilizing immunity following vaccination with live attenuated Trypanosoma cruzi parasites combined with saponin-based adjuvants”, presents a comprehensive study evaluating the use of saponin-based adjuvants (ISPA and Quil-A) in combination with live attenuated Trypanosoma cruzi (TCC) parasites to enhance vaccine safety and immune quality in a murine model. While live attenuated vaccines for T. cruzi are promising, they raise important safety concerns. The authors’ focus on saponin-based adjuvants to mitigate inflammation and promote a balanced immune response is innovative.
We thank the reviewer for acknowledging the innovative aspect of our work and for providing valuable comments for manuscript improvement.
However, the manuscript would benefit from the following clarifications and revisions:
Several sections suggest sterilizing or superior protection; however, the Results/Discussion indicate comparable protection between TCC and adjuvanted groups, with no significant differences in parasite loads post-challenge. The authors should reconcile these points and temper claims of “long-lasting protective immunity.”
The manuscript has been revised and modified to reconcile and moderate these points as suggested by the reviewer. We have also modified the title of the manuscript to reflect this view.
The methods present two different challenge doses: 100 trypomastigotes/mouse in Section 2.4 versus 1000 in Section 2.11. The correct challenge dose and timing should be clarified and reported consistently.
We thank the reviewer for pointing out this inconsistency. The correct challenge dose used throughout all the experiments was 100 trypomastigotes per mouse. The discrepancy in Section 2.11 (now 2.9) was a typographical error, which has now been corrected to ensure consistency across all relevant sections of the manuscript.
The antibody isotype and cytokine data reveal nuanced differences in immune modulation by ISPA versus Quil-A. The discussion would be strengthened by explicitly linking these observations to known adjuvant mechanisms (e.g., ISCOM-mediated antigen presentation for ISPA, or Quil-A’s influence on TH1/TH2 polarization).
We greatly appreciate this insightful observation. The Discussion has been expanded to explicitly link our findings with the known mechanisms of action of ISPA and Quil-A.
Reviewer 4 Report
Comments and Suggestions for Authors
The manuscript presents the evaluation of a vaccine candidate against T. cruzi infection, using different adjuvants or not to assess immune response, safety, and protective efficacy in a murine model. Overall, the study addresses an important and timely topic, with potentially relevant findings for the development of new immunization strategies.
The manuscript presents the evaluation of a vaccine candidate against T. cruzi infection, using different adjuvants or not to assess immune response, safety, and protective efficacy in a murine model. Below, I provide some specific comments and suggestions for improvement.
Comments on the Simple Summary
Line 19: What do you mean by underserved? Please clarify.
Line 21: "Weakened forms"; do you mean attenuated protozoa?
Line 21: What do you mean by "to train"? Please rephrase for clarity.
Line 22: Change "scientists explored" to "we evaluated".
Lines 24–29: How was immunization conducted? What do you classify as strong protection? Please include the findings that support this statement. What do you mean by refined immunity? This is confusing. Use more precise technical terms and consider rewriting this part.
General Comments on the Abstract
- Did you include a control group (e.g., PBS and attenuated protozoa)? This is mentioned in the M&M section but not clearly stated in the abstract.
- Which immune response variables did you evaluate? Please specify.
- What do ISPA, TCC, IgG, and IL stand for? Define abbreviations at first mention.
- Line 43: Replace "helped" with "improved".
Comments on the Introduction and Methods
- Line 69: Provide examples of vaccines formulated with or without specific adjuvants and present the general mean efficacy.
- Lines 74–75: Specify which antigen-presenting cells you are referring to and cite references.
- Lines 84–91: Please include references.
- Line 96: What is TCC? Define at first mention.
- Line 101: What is IFN-γ? Define at first mention.
- Line 124: Should this be "and the" or "at the"? Please correct.
Experimental Design
- Did you have 6 groups with 12 animals each, totaling 72 mice? For example:
Group 1: TCC + ISPA
Group 2: TCC + Quil-A
Group 3: TCC only
Group 4: ISPA only
Group 5: Quil-A only
Group 6: PBS control
- Please rewrite Subtopic 2.4 (M&M), as it is very difficult to follow.
- Lines 140–141: "For short-term vaccination efficacy, two weeks after the last immunization dose, 3 mice per group were euthanized..." Does this mean that 18 mice were euthanized before the first immunization? This is very confusing. Please clarify.
- Line 143: How were sera samples collected? Which type of blood collection tubes were used? How was centrifugation performed?
- Lines 145–146: Were another 18 mice euthanized? That would leave 40 animals. What happened to the others? Please clarify.
General comment on design: The experimental scheme is very difficult to follow, even with Figure 1. I strongly suggest rewriting this section.
Subtopic 2.5: How were heart and skeletal muscle tissues collected for PCR analysis? Please provide details on collection, storage, and processing. Rewrite to include methodology.
Subtopic 2.6: Same as above for histopathological analyses.
Subtopic 2.7: Were there 18 more animals? The experimental design is confusing. Please clarify.
- Lines 192–194: Were samples analyzed only by PCR? Why not histopathological or serological analyses? Please justify.
Subtopic 2.8: From where did you determine immunoglobulins; blood, serum, or plasma? Please clarify sampling procedure.
Subtopic 2.9: Were mice used to evaluate vaccine safety also used for splenocyte culture? If yes, please include this information; if not, justify.
Subtopic 2.10: Which samples were used for cytokine response; serum, plasma, or tissue protein extracts? This is unclear. Please specify.
Subtopic 2.11: How many mice were included in this analysis? The total number of animals used is unclear. Please clarify.
Subtopic 2.12: How was data distribution assessed? Were they normally or non-normally distributed? How were they normalized before ANOVA?
Comments on Results and Discussion
- The text is well presented and the results are promising, but the lack of clarity in the M&M section compromises interpretation.
- Why did you not perform statistical comparisons between vaccinated and control groups in the figures?
Figures and Tables
- Figure 1: What does the red "X" over mice indicate? In panels B and C, what was your criterion for inflammation? What exactly was observed (effector cells, inflammatory infiltrates)? Please describe in Results.
- Table 2: What about the control groups? The same applies to Figure 2 (panels C and F).
- Lines 334–347 and Figure 3: Was there interleukin production in control groups during long-term analysis? Were there significant statistical differences among groups? Please clarify.
Author Response
The manuscript presents the evaluation of a vaccine candidate against T. cruzi infection, using different adjuvants or not to assess immune response, safety, and protective efficacy in a murine model. Overall, the study addresses an important and timely topic, with potentially relevant findings for the development of new immunization strategies. The manuscript presents the evaluation of a vaccine candidate against T. cruzi infection, using different adjuvants or not to assess immune response, safety, and protective efficacy in a murine model. Below, I provide some specific comments and suggestions for improvement.
We thank the reviewer for their evaluation of our manuscript and we are grateful for their recognition that our study could provide a relevant approach for new immunization strategies.
Comments on the Simple Summary
Line 19: What do you mean by underserved? Please clarify.
Line 21: "Weakened forms"; do you mean attenuated protozoa?
Line 21: What do you mean by "to train"? Please rephrase for clarity.
Line 22: Change "scientists explored" to "we evaluated".
Lines 24–29: How was immunization conducted? What do you classify as strong protection? Please include the findings that support this statement. What do you mean by refined immunity? This is confusing. Use more precise technical terms and consider rewriting this part.
According to the journal guidelines, a simple summary should be a concise and easily understandable overview of the main content of a research article, intended for an audience beyond specialists in the field. Nonetheless, it has been entirely revised and re-writed according to the reviewer comments (and limited word counting) to include more precise and technical terminology.
General Comments on the Abstract
- Did you include a control group (e.g., PBS and attenuated protozoa)? This is mentioned in the M&M section but not clearly stated in the abstract.
This information is stated in the abstract.
- Which immune response variables did you evaluate? Please specify.
We have modified the Abstract according to the reviewer suggestion.
- What do ISPA, TCC, IgG, and IL stand for? Define abbreviations at first mention.
Modified according to the reviewer suggestion.
- Line 43: Replace "helped" with "improved".
Modified according to reviewer suggestion.
Comments on the Introduction and Methods
- Line 69: Provide examples of vaccines formulated with or without specific adjuvants and present the general mean efficacy.
Modified according to the reviewer suggestion.
- Lines 74–75: Specify which antigen-presenting cells you are referring to and cite references.
Modified according to the reviewer suggestion.
- Lines 84–91: Please include references.
The reference corresponding to that paragraph was already included in the manuscripti (Bertona et al.). Besides, we have added additional complementary references at the end of the paragraph.
- Line 96: What is TCC? Define at first mention.
Corrected as previously suggested.
- Line 101: What is IFN-γ? Define at first mention.
Corrected as previously suggested.
- Line 124: Should this be "and the" or "at the"? Please correct.
Modified.
Experimental Design
- Did you have 6 groups with 12 animals each, totaling 72 mice? For example:
Group 1: TCC + ISPA
Group 2: TCC + Quil-A
Group 3: TCC only
Group 4: ISPA only
Group 5: Quil-A only
Group 6: PBS control
The reviewer is correct, this is specified in section 2.4 (M&M) and groups are also represented in Figure 1 to facilitate reader's comprehension.
- Please rewrite Subtopic 2.4 (M&M), as it is very difficult to follow.
- Lines 140–141: "For short-term vaccination efficacy, two weeks after the last immunization dose, 3 mice per group were euthanized..." Does this mean that 18 mice were euthanized before the first immunization? This is very confusing. Please clarify.
- Line 143: How were sera samples collected? Which type of blood collection tubes were used? How was centrifugation performed?
- Lines 145–146: Were another 18 mice euthanized? That would leave 40 animals. What happened to the others? Please clarify.
General comment on design: The experimental scheme is very difficult to follow, even with Figure 1. I strongly suggest rewriting this section.
The entire Section 2.4, along with the corresponding figure illustrating the experimental design, was revised incorporating the reviewer’s suggestions and comments.
Subtopic 2.5: How were heart and skeletal muscle tissues collected for PCR analysis? Please provide details on collection, storage, and processing. Rewrite to include methodology.
Modified according to reviewer suggestion.
Subtopic 2.6: Same as above for histopathological analyses.
This information is already in the manuscript.
Subtopic 2.7: Were there 18 more animals? The experimental design is confusing. Please clarify.
Yes, the reviewer is correct. We agree that the distinction between the experimental procedures was not entirely clear. To improve the clarity of the manuscript, we have now moved the description of this separate experiment and its methods to a dedicated Section 2.10.
- Lines 192–194: Were samples analyzed only by PCR? Why not histopathological or serological analyses? Please justify.
The experiment referred to was conducted as an additional control, solely to determine whether the attenuated parasites used for vaccination could be detected by a highly sensitive method (qRT-PCR) in animals immunosuppressed with cyclophosphamide. From our perspective, histopathological evaluation would not have provided meaningful information, since detecting amastigote nests by this technique is much more difficult, also damage markers we might have observed could be related to the immunosuppression itself. Furthermore, it would not have been appropriate to measure antibody levels in those animals, as cyclophosphamide reduces antibody production by inhibiting B and T cell activity.
Subtopic 2.8: From where did you determine immunoglobulins; blood, serum, or plasma? Please clarify sampling procedure.
Modified according to reviewer requirement (section 2.5 of revised manuscript).
Subtopic 2.9: Were mice used to evaluate vaccine safety also used for splenocyte culture? If yes, please include this information; if not, justify.
This information was stated in lines 206 and 207 of the original manuscript (section 2.6 of revised manuscript).
Subtopic 2.10: Which samples were used for cytokine response; serum, plasma, or tissue protein extracts? This is unclear. Please specify.
This information was specified in line 216-217 of the original manuscript. However, for a better comprehension all the information is included in one section (section 2.6 of revised manuscript).
Subtopic 2.11: How many mice were included in this analysis? The total number of animals used is unclear. Please clarify.
This information is now incorporated in sections 2.5 and 2.9 (new version of the manuscript).
Subtopic 2.12: How was data distribution assessed? Were they normally or non-normally distributed? How were they normalized before ANOVA?
Data distribution was assessed using the Shapiro–Wilk normality test. The results indicated that the datasets followed a normal distribution, and therefore no further normalization was required prior to applying ANOVA. This information has now been incorporated into the manuscript (Section 2.11 ).
Comments on Results and Discussion
- The text is well presented and the results are promising, but the lack of clarity in the M&M section compromises interpretation.
We thank the Reviewer's observation. According to their comment, the section Material and Methods was modified for a better comprehension.
- Why did you not perform statistical comparisons between vaccinated and control groups in the figures?
All statistical comparisons were performed against the TCC control group (without adjuvants), as this is the reference group we are interested in assessing for potential differences. The other experimental groups serve as additional controls to rule out any nonspecific background effects that could compromise our results. Specifically, our main objective is to determine whether the adjuvants enhance the performance of TCC and safety of the TCC vaccine, which is why all comparisons were made with this control group.
Figures and Tables
- Figure 1: What does the red "X" over mice indicate? In panels B and C, what was your criterion for inflammation? What exactly was observed (effector cells, inflammatory infiltrates)? Please describe in Results.
The red cross means that animals were sacrificed. This information was added to the figure legend. Also, criterion for inflammation was described in the Results section.
- Table 2: What about the control groups? The same applies to Figure 2 (panels C and F).
Since the experiment objective was to assess whether immunizing parasites could be detected in immunosuppressed animals, non-immunized control groups were omitted as in such cases the qRT-PCR results are inherently negative.
For Figures 2C and 2F, the plotted data represents the IgG1/IgG2 ratio of anti-T. cruzi antibodies. As the control groups were not immunized, their specific serology was negative. Consequently, their absorbance values were zero, and the ratios were not calculable for these groups.
- Lines 334–347 and Figure 3: Was there interleukin production in control groups during long-term analysis? Were there significant statistical differences among groups? Please clarify.
No cytokine production was detected in the control groups at the long term condition (Figure 4 now). Significant differences are indicated in the figure when present, and in cases where no differences are observed, this is also marked.
Round 2
Reviewer 1 Report
Comments and Suggestions for Authors
Thank you for making the suggested modifications to the manuscript. All in all, I agree with the authors that this is a good start and preliminary evaluation of a live attenuated vaccine for Chagas disease. I have no further changes to suggest.
Reviewer 2 Report
Comments and Suggestions for Authors
The authors are respond to each comment clearly, respectfully, and with specific details according to the suggestions
Comments on the Quality of English LanguageThe English language used in the paper is clear, grammatically sound, and easily understandable. The writing effectively conveys the intended meaning without ambiguity, making the manuscript accessible to a broad academic audience. Overall, the language quality supports the clarity and professionalism of the work